# Methylation of SPARCL1 Is Associated with Oncologic Outcome of Advanced Upper Urinary Tract Urothelial Carcinoma

**DOI:** 10.3390/ijms20071653

**Published:** 2019-04-03

**Authors:** Hao-Lun Luo, Po-Huang Chiang, Chun-Chieh Huang, Yu-Li Su, Min-Tse Sung, Eing-Mei Tsai, Chang-Shen Lin, Po-Hui Chiang

**Affiliations:** 1Graduate Institute of Medicine, College of Medicine, Kaohsiung Medical University, Kaohsiung 83301, Taiwan; alesy1980@gmail.com (H.-L.L.); csl@kmu.edu.tw (C.-S.L.); 2Department of Urology, Kaohsiung Chang Gung Memorial Hospital and Chang Gung University College of Medicine, Kaohsiung 83301, Taiwan; 3Institute of Population Health Sciences, National Health Research Institutes, Miaoli 35053, Taiwan; chiangp@nhri.org.tw; 4Department of Health Risk Management, College of Management, China Medical University, Taichung 40402, Taiwan; 5Institute of Biomedical Informatics, National Yang Ming University, Taipei 11221, Taiwan; 6Department of Radiation Oncology, Kaohsiung Chang Gung Memorial Hospital and Chang Gung University College of Medicine, Kaohsiung 83301, Taiwan; cgukinace@gmail.com; 7Department of Hematology and Oncology, Kaohsiung Chang Gung Memorial Hospital and Chang Gung University College of Medicine, Kaohsiung 83301, Taiwan; yolisu@mac.com; 8Department of Pathology, Kaohsiung Chang Gung Memorial Hospital and Chang Gung University College of Medicine, Kaohsiung 83301, Taiwan; mtsmts@cgmh.org.tw; 9Department of Obstetrics and Gynecology, Kaohsiung Medical University Hospital, Kaohsiung Medical University, Kaohsiung 83301, Taiwan; 10Graduate Institute of Medicine, Center of Excellence for Environmental Medicine, Kaohsiung Medical University, Kaohsiung 83301, Taiwan; 11Department of Biological Sciences, National Sun Yat-sen University, Kaohsiung 83301, Taiwan

**Keywords:** SPARCL1, upper urinary tract urothelial carcinoma

## Abstract

Advanced upper urinary tract urothelial carcinoma (UTUC) is often associated with poor oncologic outcomes. The secreted protein acidic and rich in cysteine-like 1 (SPARCL1) protein, belongs to the SPARC-related family of matricellular proteins. Much literature has been published describing the role of SPARCL1 in the prognosis many cancers. In this study, methylated promoter regions in high-grade and high-stage upper urinary urothelial tumours compared with normal urothelium were analyzed and revealed that *SPARCL1* was the most significantly hypermethylated gene in UTUC tissues. Then we prospectively collected UTUC samples and adjacent normal urothelium for pyrosequencing validation, identifying significant CpG site methylation in UTUC tissues. In addition, SPARCL1 RNA levels were significantly lower in UTUC samples. Multivariate Cox regression analysis from 78 patients with solitary renal pelvic or ureteral pT3N0M0 urothelial carcinomas revealed that only negative SPARCL1 expression and nonpapillary tumour architecture were independently associated with systemic recurrence (*p* = 0.011 and 0.008, respectively). In vitro studies revealed that the behaviour of BFTC-909 cells was less aggressive and more sensitive to radiation or chemotherapy after SPARCL1 overexpression. Thus, SPARCL1 could be considered as a prognostic marker and help decision-making in clinical practice.

## 1. Introduction

Upper urinary tract urothelial carcinoma (UTUC) is a relatively rare disease compared with urinary bladder urothelial carcinoma. Advanced UTUC is often associated with poor oncologic outcome [1]. Chemotherapy is the current standard therapy for advanced UTUC in the neoadjuvant or adjuvant setting [2]. However, the high prevalence of renal insufficiency in UTUC is a clinical challenge [3]. For those patients with renal insufficiency, which means they are ineligible for chemotherapy, radiation therapy is sometimes an adjunct strategy. Further exploration of treatment targets sensitive to treatment multimodality may help improve clinical outcome.

The secreted protein acidic rich in cysteine-like 1 (SPARCL1) protein belongs to the SPARC-related family of matricellular proteins. The human *SPARCL1* gene was initially discovered in high endothelial venules from tonsils [4]. Several studies have described the role of SPARCL1 in the prognosis of colorectal, gastric, ovarian, and prostate cancers [5,6,7,8]. The function of SPARCL1 is not completely known, but studies have suggested that it may modulate high endothelial cell adhesion to the basement membrane as an antiadhesive protein [9]. Additionally, SPARCL1 was reported to inhibit the progression of cells from G_1_ to S phase and to negatively regulate cell proliferation [10]. It is also a tumour suppressor as it induces cell differentiation possibly via MET, which represses the aggressiveness of CRCs [5].

To the best of our best knowledge, no studies have yet reported the role of SPARCL1 in urothelial carcinoma. We found that high-stage/high-grade UTUC samples had significant *SPARCL1* hypermethylation compared with normal urothelium adjacent to low-stage/low-grade specimens. The objective of this study was to identify the role of SPARCL1 in advanced UTUC.

## 2. Results

### 2.1. DNA Hypermethylation Genes in UTUC

To explore potential cancerous factors in UTUC, we first compared DNA methylation in high-grade and high-stage urothelial tumour samples with that in normal urothelium samples by methyl-CpG binding domain (MBD) protein capture for genome-wide DNA methylation analysis. The differentially methylated promoter regions in three high-grade and high-stage urothelial tumour samples compared with three normal urothelium samples are shown in Figure 1A. A principal component analysis plot showed that the tumour and normal tissues were quite distinguishable based on differences in DNA methylation (Figure 1B). The top 10 hypermethylated genes are shown in Figure 1C; SPARCL1 hypermethylation was found to be increased 20-fold in UTUC tissue (*p* = 4 × 10^−5^). It has also been reported that DNA methylation is the reason for the downregulation of *SPARCL1* and demethylation of the gene partially reversed the abnormal expression in pancreatic cancer [11] and osteosarcoma [12]. However, it is not known whether SPARCL1 has any role in the development of UTUC. Therefore, we set out to study the role of SPARCL1 in human UTUC tissue.

### 2.2. SPARCL1 Promoter Hypermethylation and Attenuated Expression in UTUC

We next validated the DNA hypermethylation of *SPARCL1* in UTUC (Figure 2A). *SPARCL1* had differentially methylated region starting from −1129 from the start codon. We used target-specific sequencing at the *SPARCL1* promoter region via pyrosequencing and identified 4 methylation sites (Figure 2B). The methylation status of four CpG site from 25 paired UTUC and adjacent urothelium is shown in Figure 2C,D. The results identified significant CpG site hypermethylation in UTUC tissue compared with normal urothelium. Consequently, we further evaluate the mRNA levels of *SPARCL1* in 55 pairs of UTUC tissues and their matched normal tissues. The UTUC samples were found to have significantly lower *SPARCL1* RNA expression through real-time PCR (Figure 2E). Taken together, these results demonstrated that *SPARCL1* DNA hypermethylation was increased and *SPARCL1* mRNA expression was decreased in UTUC.

### 2.3. Low SPARCL1 Expression Is Associated with Advanced UTUC Stage and More Distant Metastasis in Retrospective TMA Cohort

To assess the expression of SPARCL1 protein in UTUC, we evaluated the status of SPARCL1 by tissue microarrays (Figure 3A and Appendix A). The expression of SPARCL1 in pathologic tumor stages 3 and 4, patients were significantly lower than that in pathologic tumor stages 0, 1, and 2, patients on the basis of a comparison of immunoreactivity score (Figure 3B and Appendix A). Our data showed that the expression level of SPARCL1 was downregulated in M1 tumours compared with M0 tumours (Figure 3C and Appendix A). The characteristics of included patients are listed in Table 1. The clinical and pathological factors were equal in both groups, but significantly more systemic disease recurrence was found in the negative SPARCL1 expression group (*p* = 0.042). Multivariate Cox regression analysis revealed that only negative SPARCL1 expression and nonpapillary tumour architecture were independently associated with systemic recurrence compared with currently known prognostic factors (*p* = 0.011 and 0.008, hazard ratios = 2.89 and 3.01, respectively) (Table 2). A Kaplan–Meier plot showed that patients with negative SPARCL1 expression commonly developed systemic recurrences (*p* = 0.033) (Figure 3D). The difference was clear, especially 1 year after surgical intervention. In addition, real-time PCR and western blot analyses of the high grade urothelial carcinoma cell lines (J82, BFTC909, and T24) displayed lower *SPARCL1* expression levels than the low grade urothelial carcinoma cell line (RT4) and the immortalized human urothelial cells (SV-HUC-1) in Figure 3E,F. Methylation validation in urothelial carcinoma cell lines revealed that *SPARCL1* methylation levels for all 4 of the CpG sites were significantly higher in J82, BFTC909, and T24 cell lines than in RT4 cell line (Figure 3G). To further examine the methylation-mediated downregulation of *SPARCL1* in UC, we treated SV-HUC-1, RT4, and BFTC909 cell lines with a specific methyltransferase inhibitor, 5-aza-2′-deoxycytidine. The *SPARCL1* mRNA expression in UC cell lines was increased by 5-aza-2′-deoxycytidine treatment (Figure 3H). 

### 2.4. SPARCL1 Regulates the Cell Proliferation and Migration in Vitro and Improves the Anti-Tumour Effect of Cisplatin or Radiation Treatment

The decreased expression of SPARCL1 in UTUC clinical samples led us to investigate whether endogenous SPARCL1 plays a role in growth capability of UTUC cell lines. In vitro UTUC behaviour revealed that the UTUC cell line (BFTC909) weakly expressed SPARCL1, which is compatible with its aggressive tumour behaviour. SPARCL1 overexpression was induced in the BFTC909 cell line, following which colony assay was performed, indicating that overexpression of SPARCL1 in vitro significantly decreased cell viability (Figure 4A). A previous report similarly showed SPARCL1 suppresses the proliferation and migration of human ovarian cancer [13]. Therefore, we investigated whether SPARCL1 plays a role in metastasis of UTUC. The migration of BFTC909 cells was significantly enhanced by SPARCL1 knockdown, while overexpression of SPARCL1 inhibited cell migration (Figure 4B), indicating that the tumour behaviour of BFTC909 cells was less aggressive after SPARCL1 overexpression. Epithelial–mesenchymal transition (EMT) is known to play an important role in cancer progression and metastasis. Next, we examined the correlation between SPARCL1 expression and expression of EMT markers. These data showed that SPARCL1 overexpression significantly reduced expression of N-cadherin and vimentin and induced E-cadherin expression (Figure 4C). In addition, we investigated whether combined treatment with SPARCL1 and ionizing radiation inhibited clonogenicity in vitro. Radiation alone (0, 2, 4, 6, or 8 Gy) or in combination with SPARCL1 overexpression restrained clonogenic formation in BFTC909 and the combined treatment had a synergistic anti-tumor effect (Figure 4D). Moreover, we observed that SPARLC1 overexpression enhanced anti-tumor effect of cisplatin treatment, as presented in Figure 4E. Collectively, the results indicated that SPARCL1 suppresses proliferation and migration in vitro and enhances the anti-tumor effect of cisplatin treatment or radiation therapy.

## 3. Discussion

Advanced UTUC is often associated with early local regional recurrence or distant metastasis, leading to poor survival outcomes even after standard radical nephroureterectomy [1]. Therefore, the treatment strategy needs to include perioperative chemotherapy to improve survival [2]. However, perioperative renal insufficiency, especially after radical surgery, was observed in patients with UTUC. Such a high prevalence of renal function impairment leads to an inadequate dose of cisplatin-based chemotherapy [3]. Though an immune checkpoint inhibitor was developed as an alternative treatment choice for patients with urothelial carcinoma ineligible for cisplatin therapy, the response rate was still limited [14]. Urothelial carcinoma is not always curable if distant metastasis develops. Moreover, urinary bladder cancer recurrence is not uncommon after UTUC management, owing to the multifocal characteristics of urothelial carcinoma [15]. Potential advanced bladder cancer development often causes major disability in these patients after radical surgery [16]. Further investigation into tumorigenesis is needed to improve cancer control and prevent further metastasis of UTUC.

DNA methylation is the C5 methylation of cytosine bases in a CpG dinucleotide. DNA methylation serves as an epigenetic mark to organize the complex human genome. In cancers, DNA methylation patterns are generally disturbed compared with the corresponding normal tissues. DNA hypermethylation of CpG island promoters is commonly associated with transcriptional repression of the affected promoter, leading to decreased gene expression or alternative promoter use [17]. Therefore, the hypermethylation of tumour suppressor genes may be associated with tumorigenesis due to epigenetic change. Such methods have been investigated mostly in urinary bladder urothelial carcinoma by studying tumour or urine samples to serve as prognostic biomarkers or diagnostic tools for early detection [18]. The application of DNA methylation in UTUC is less frequently reported, owing to its relatively rare incidence. In Taiwan, an unusually high prevalence of UTUC has been reported, and it is a major public health problem [19]. The study of DNA methylation with adequate validation cohorts may help in preventing or detecting UTUC.

To identify any epigenetic change of the tumorigenesis of UTUC, we selected three tumours of high-grade and high-stage UTUC and three normal urothelial tissue samples adjacent to low-grade and low-stage UTUC by comparing the methylation intensities of promoter regions. The methylation of *SPARCL1* was significantly greater in UTUC than in normal urothelium. Pyrosequencing-based analysis of our prospectively collected UTUC and normal urothelial tissue samples further proved significant *SPARCL1* hypermethylation in UTUC. Such a high prevalence of *SPARCL1* hypermethylation in the UTUC samples indicates that the loss of SPARCL1 function might be considered to have clinical utility in the assessment of UTUC behavior. In this study, the 5-year recurrence-free survival of advanced UTUC was approximately 60% in the SPARCL1-positive group, whereas it was approximately 35% in the SPARCL1-negative group (presented as Kaplan Meier plot in Figure 3D). Both SPARCL1 presentation and tumour architecture were significant in regard to systemic UTUC recurrence in our institutional cohort.

The prognostic value of SPARCL1 and tumour architecture is independent in multivariate analysis, compared to other currently known prognostic factors. The BFTC909 cell line originates from an aggressive renal pelvic tumour with low SPARCL1 expression in Taiwan. Our in vitro study shows that its aggressive behaviour can be reversed by SPARCL1 overexpression in colony formation assays and migration assays. A lower N-cadherin/E-cadherin ratio, which indicates less epithelial–mesenchymal transition and cell migration, is also compatible with less malignant behaviour after SPARCL1 overexpression. The sensitivity to radiation and chemotherapy can be also observed in SPARCL1 overexpressing cells. Further correlation studies between SPARCL1 expression and treatment response for locally advanced UTUC might provide more information in the future. Therefore, from the preliminary result, SPARCL1 could be considered a prognostic biomarker for advanced UTUC and further identification of the outcome of the clinical benefit of chemotherapy or radiation on patients with positive SPARCL1 UTUC needs further clinical trial for validation of its prognostic utility.

The role of SPARCL1 is commonly discussed in gastrointestinal malignancies. A meta-analysis of eight studies including a total of 2356 patients revealed that its expression is associated with less lymph node and distant metastasis. The negative association between the SPARCL1 expression and poor tumour differentiation indicates the major role of aggressive tumour behavior [20]. In addition, the prognostic impact of SPARCL1 expression in ovarian and prostate cancers has also been reported [7,8]. The mechanism of SPARCL1 in cancer behaviour in these studies involved cell growth, proliferation, differentiation, cell cycle inhibition, and regulation of the microenvironment [9,10,11]. We examined the status of SPARCL1 methylation status of several UC cell lines and found that SPARCL1 methylation was more common in aggressive UC cell lines. However, to the best of our knowledge, no report on the prognostic role of SPARCL1 in urothelial carcinoma has been published to date. On the basis of the prospective tissue analysis and retrospective cohort validation in this study, we believe that the prognostic role of SPARCL1 expression should not be overlooked in UTUC.

This is the first study on the impact of SPARCL1 expression on advanced UTUC. In this study, we demonstrated that SPARCL1 may have clinical utility as a prognostic biomarker that is independently associated with UTUC recurrence. The major limitation is its retrospective single-institution design with limited case numbers and only a Taiwanese population represented. However, we selected a retrospective advanced UTUC cohort in this study with a primary focus on pT3 disease, instead of pT4 disease, because these patients can often be treated with surgical resection for representative pathologic review. Unlike low-stage disease, pT3 UTUC often recurs within 2 years, and the average follow-up duration in this study is long enough to observe the oncologic outcome. The prognostic effect of biomarkers in low-stage UTUC may interfere with the curative effect radical surgery. In addition, we carefully excluded perioperative chemotherapy in order to observe natural UTUC behaviour. The endpoint of this study was systemic disease recurrence instead of patient survival because palliative or salvage therapy after systemic disease recurrence might add more confounding factors to the analysis of the prognostic role of biomarkers. In addition, the methylation status of cell lines in this study are not all UTUC. To better understand the methylation status between the aggressive and non-aggressive cell lines, the methylation status in primary culture cells should be validated.

The significant correlation between SPARCL1 expression and advanced UTUC behaviour is an interesting finding. In this study, careful selection of a retrospective advanced UTUC cohort, and in vitro cancer behaviour analysis proved the important role of SPARCL1 in UTUC. The conclusion about the oncologic impact of SPARCL1 is similar in many other cancer studies. Studies on the prevalence of SPARCL1 expression in UTUC in patients of other races are still needed to clarify the role of application in clinical practice worldwide. In Taiwan, *SPARCL1* methylation is commonly found in UTUC. Further therapeutic agents modulating *SPARCL1* methylation or development of SPARCL1-based screening may help with UTUC treatment or prevention.

## 4. Materials and Methods

### 4.1. Study Patient Selection

First, we started to collect the prospective UTUC samples since 2016. We included 6 patients for DNA methylation analysis. Then 25 patients for DNA methylation validation (pyrosequencing) and 55 patients using real-time PCR detected SPARCL1 are from prospectively collected samples (Appendix A). The relationship between SPARCL1 expression and pathological stage/distant metastasis in retrospective whole stage distribution TMA cohort (Appendix A). However, for clinical translation, we want to identify the clinical prognostic role of SPARCL1 for advanced UTUC. Therefore, the IHC information from 78 patients for clinical advanced UTUC outcome cohort is from the full representative slide by pathologist’s review. The 78 patients with solitary ureteral or renal pelvic pT3N0M0 urothelial carcinomas without perioperative neoadjuvant or adjuvant therapy were included in the study. Pathological features and clinical outcome assessment were as previously described [21]. This study was approved by Kaohsiung Chang Gung Medical Center Institutional Review Board since 01/Mar/2016 (approval number 106-4117C). Informed consent was obtained preoperatively.

### 4.2. DNA Methylation Analysis

DNA from the clinical tissues was isolated using the QIAamp DNA Mini Kit (Qiagen, Hilden, Germany). After passing the quality control criteria, the DNA samples were first subjected to immunoprecipitation using proteins with a methyl-CpG-binding domain. Then, the enriched DNA fragments were amplified through polymerase chain reaction (PCR) and loaded onto the GeneChip Human Promoter 1.0R tiling array (Affymetrix, Santa Clara, CA, USA). We determined the difference in methylation between tumour samples of high-grade and high-stage UTUC (*n* = 3) and normal urothelium adjacent to low-grade and low-stage UTUC (*n* = 3) by comparing the probe intensities of promoter regions.

### 4.3. Pyrosequencing-Based Bisulfite PCR Analysis

A total of 500 ng of DNA from each sample was treated using the EZ DNA Methylation-Lightning Kits Bisulfite Conversion System (Zymo Research, Irvine, CA, USA) and the converted DNA was eluted in 20 μL of the elution buffer. Bisulfite conversion was performed in the dark at 98 °C for 10 min and 64 °C for 3.5 h, followed by desulphonation of the converted DNA. Gene amplification was performed using the HotStarTaq^®^ Master Mix Kit (Qiagen). The CpG1, CpG2, and CpG3 sites of SPARCL1 pyrosequencing primers used were as follows: forward: 5′-GGTGTGTGGGAAAAGTTTTAGAT-3′; reverse: 5′-CCAAATTTCCAATTTCTCTTAAACC-3′; sequencing: 5′-TTATTTAATTTTTTTGAGTTTTAT-3′. The CpG4 of SPARCL1 pyrosequencing primers used were as follows: forward: 5′-GTGGGAAAAGTTTTAGATTTAGAGTT -3′; reverse: 5′-CCAAATTTCCAATTTCTCTTAAACC-3′; sequencing: 5′-AAAAGTGAGATAGATTAAGTATA-3′. Amplification conditions were as follows: 95 °C for 15 min, 94 °C for 30 s, 60 °C for 30 s, 72 °C for 30 s, 45 cycles. To assay DNA methylation levels of the SPARCL1 promoter, bisulfite sequencing was performed on the PyroMark Q24 instrument (Qiagen). Relative levels of methylation at each CpG site were analyzed with PyroMark Q24 version 2.0.6 software.

### 4.4. Immunohistochemistry and Patient Grouping

A human UTUC TMA containing 103 specimens (with triplicate cores for each sample) was provided from Chang Gung Medical Foundation Kaohsiung Chang Gung Memorial Hospital Tissue Bank. Immunostaining for SPARCL1 was performed on a fully automated Bond-Max system (Leica Microsystems, Wetzlar, Germany). Slides carrying tissue sections cut from formalin-fixed, paraffin-embedded tissue microarray blocks were dried for 1 hour at 60 °C. These slides were then covered by Bond Universal Covertiles and placed into the Bond-Max instrument. All subsequent steps were performed automatically by the instrument according to the manufacturer’s instructions (Leica Microsystems) according to a previous report [21]. The following procedure was used: (1) Deparaffinization of tissue on slides by rinsing with Bond Dewax Solution at 72 °C; (2) heat-induced epitope retrieval (antigen unmasking) with Bond Epitope Retrieval Solution 2 for 10 min at 100 °C; (3) peroxide block placement on the slides for 10 min at room temperature; (4) incubation with a mouse monoclonal anti-SPARCL-1 antibody at a dilution of 1:100 for 60 min at room temperature; (5) incubation with Post Primary reagent for 10 min at room temperature; (6) Bond Polymer placement on the slides for 8 min at room temperature; (7) color development with 3,3′-diaminobenzidine tetrahydrochloride (DAB) as a chromogen for 3 min at room temperature; and (8) hematoxylin counterstaining for 1 minutes. Slides were mounted and examined by light microscopy.

The immunoreactivity scoring was based on the intensity of positive staining using a 3-point scale: 0–10%, 0; 11–50%, 1; 51–80%, 2; and >80%, 3. The identification of SPARCL1 expression was performed by a uropathologist (Dr. Min-Tse Sung) and urooncologist (Dr. Hao Lun Luo) in our institution.

### 4.5. RNA Isolation and Real-Time PCR

RNA was extracted by means of QIAGEN RNA purification kit from UTUC clinical tissue and cell lines. Five microgram RNA of each sample will be reverse transcribed using RevertAid^TM^ H Minus Reverse Transcriptase (Fermentas, Waltham, MA, USA). Real-time PCR was performed using SYBR Green PCR master mix (Life Technologies, Carlsbad, CA, USA) and ABI 7500 sequence detection system (Life Technologies). The real-time PCR primers: *SPARCL1* forward: 5′-GTGAAGGCAACATGAGGGTGCA-3′; *SPARCL1* reverse primer: 5′-GTTGGAGGACAAGTCACTGGATC-3′. *GAPDH* forward: 5′-GTCTCCTCTGACTTCAACAGCG-3′; *GAPDH* reverse primer: 5′-ACCACCCTGTTGCTGTAGCCAA-3′. All primers were purchased from OriGene (Rockville, MD, USA) and checked for specificity using BLAST (NCBI). Exon/intron junctions were spanned.

### 4.6. Western Blotting Assays

Cells were lysed directly in a modified RIPA buffer containing a protease inhibitor mixture (Roche, Basel, Switzerland). The relative protein concentration in the supernatants was related to standard BSA concentrations determined by BCA protein assay kit (Thermo Fisher Scientific, Waltham, MA, USA). For each lane of 8~10% SDS–PAGE gel, 40 ug protein of cell lysates were loaded, separated and subsequently transferred onto Immobilon-P Transfer Membrane (Millipore, Burlington, MA, USA). The membranes were probed with specific antibodies. The primary antibodies were against SPARCL1 (Abcam, Cambridge, MA, UK), Vimentin (Cell Signaling Technology, Danvers, MA, USA), N-cadherin (Cell Signaling Technology, Danvers, MA, USA), E-cadherin (Cell Signaling Technology, Danvers, MA, USA), β-actin (Chemicon, Temecula, CA) and GAPDH (GeneTex, Irvine, CA, USA). The primary antibodies were used: SPARCL1 (1:500), Vimentin (1:1000), N-cadherin (1:1000), E-cadherin (1:1000), β-actin (1:5000) and GAPDH (1:5000). The secondary antibodies were added and incubated for 2 h and then visualized using chemiluminescence. Enhanced chemiluminescence (ECL) western blotting reagents were obtained from Pierce Biotechnology (Rockford, IL, USA).

### 4.7. Cell Line Study

SV-HUC-1, RT4, J82, BFTC909, and T24 cell lines were purchased from Bioresource Collection and Research Center (BCRC). The plasmid expressing shSPARCL1 and the cDNAs coding SPARCL1 wild-type were purchased from OriGene. All constructs were verified by DNA sequencing analyses. Plasmids were isolated by QIAGEN Plasmid Mini Kit and transfection was performed by using TurboFect transfection reagents (Thermo Fisher Scientific, Waltham, MA, USA) according to the manufacturer’s instructions. The migration assay we used has been described in detail previously [21]. SV-HUC-1, RT4, and BFTC909 cells were treated with vehicle or 2, 5, 10 μM 5-Aza-2′-deoxycytidine (Sigma-Aldrich, Saint Louis, MO, USA) for 2 days. Media with vehicle or 5-Aza-2′-deoxycytidine was changed daily. The histology of UTUC is urothelial carcinoma. The included cell lines RT4/J82/T24 are bladder urothelial carcinomas. However, we can only obtain the aggressive UTUC cell line (BFTC 909) from Taiwan instead of non-aggressive UTUC cell line (actually all the available UTUC cell lines in the world are aggressive in behavior, which is compatible with the finding from clinical practice). Therefore, we examined other UC cell lines to identify the SPARCL methylation in the aggressive UC cell line (BFTC909/T24/J82) and relatively non-aggressive UC cell line (RT4).

### 4.8. Colony Formation Assay

Cells of different conditions (Control or SPARCL1 knockdown/overexpression). The BFTC909 cells were seeded in the 25T flasks. For the clonogenic assay, 0, 2, 4, 6, and 8 Gy were delivered using 6MV photon beam from the linear accelerator. They were irradiated from one posterior-anterior portal with 1cm bolus. We seeded different cell number (50–7200) according to the radiation doses in six-well plates. After 13 days, we used methanol to fix cells and 0.5% crystal violet to stain colonies and used distilled water wash twice. Cells were counted using a stereomicroscope.

### 4.9. Statistical Analysis

SPSS Statistics version 17 software (SPSS, Chicago, IL, USA) was used for all statistical analyses. The chi-squared test and independent samples *t*-test were used for inter-group comparisons. The Kaplan–Meier method with the log-rank test was used for time-to-event analysis. Multivariate Cox regression analysis was used to assess the independent roles of perioperative factors in systemic recurrence [21]. A *p* value less than or equal to 0.05 was used to define a statistically significant result. The observed endpoint in this study was systemic recurrence before initiation of systemic anticancer therapy.

## 5. Conclusions

We found that high-stage/high-grade UTUC samples had significant SPARCL1 hypermethylation compared with normal urothelium adjacent to low-stage/low-grade specimens. In addition, positive SPARCL1 expression was an independent prognostic biomarker for locally advanced UTUC in our clinical cohort. Less malignant UTUC behaviour and increased sensitivity to radiation and chemotherapy is observed in vitro study.

## Figures and Tables

**Figure 1 ijms-20-01653-f001:**
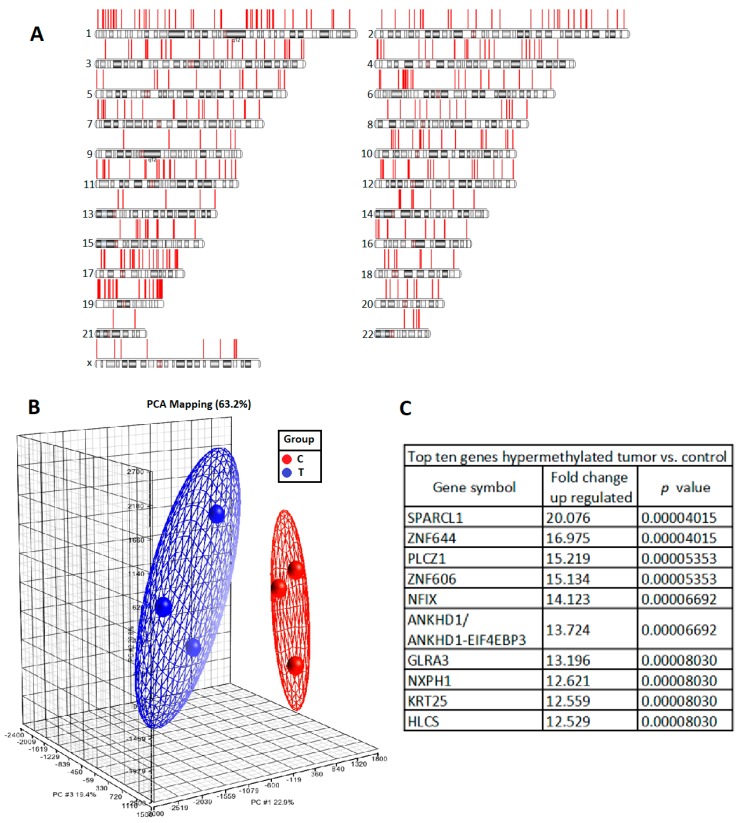
DNA methylation profiles between the high-grade and high-stage urothelial tumor and normal urothelium sets. We conducted methylation microarray assays on 3 UTUC and 3 normal urothelium samples and the raw data generated was analyzed with Partek. (**A**) Hypermethylated promoter region of each chromosome. (**B**) Principal component analysis plot based on methylation profiles in UTUC (red) and normal urothelium (blue). (**C**) The top-10 hypermethylated genes in UTUC compared with normal urothelium samples.

**Figure 2 ijms-20-01653-f002:**
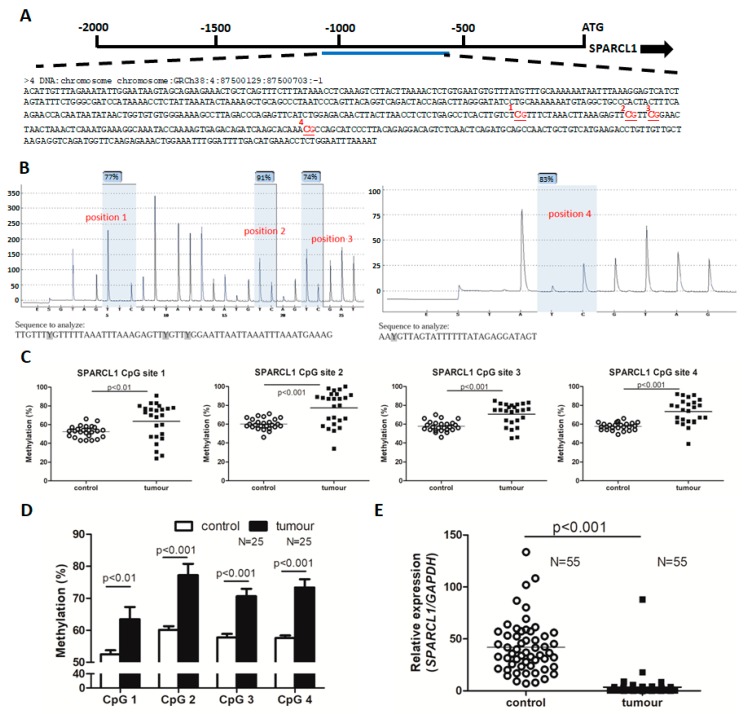
*SPARCL1* associated with hyper-DMR was down-regulated in the upper urinary tract urothelial carcinoma. (**A**) Schematic view of the *SPARCL1* locus. There were four CpG sites in the *SPARCL1* promoter region. (**B**) Four methylation sites in the SPARCL1 promoter region identified by bisulphite pyrosequencing. (**C**,**D**) *SPARCL1* CpG site methylation levels in the control (adjacent urothelium, *n* = 25) and the paired tumor samples. (**E**) Quantitative polymerase chain reaction from UTUC (*n* = 55 patients) and their matched adjacent normal tissues.

**Figure 3 ijms-20-01653-f003:**
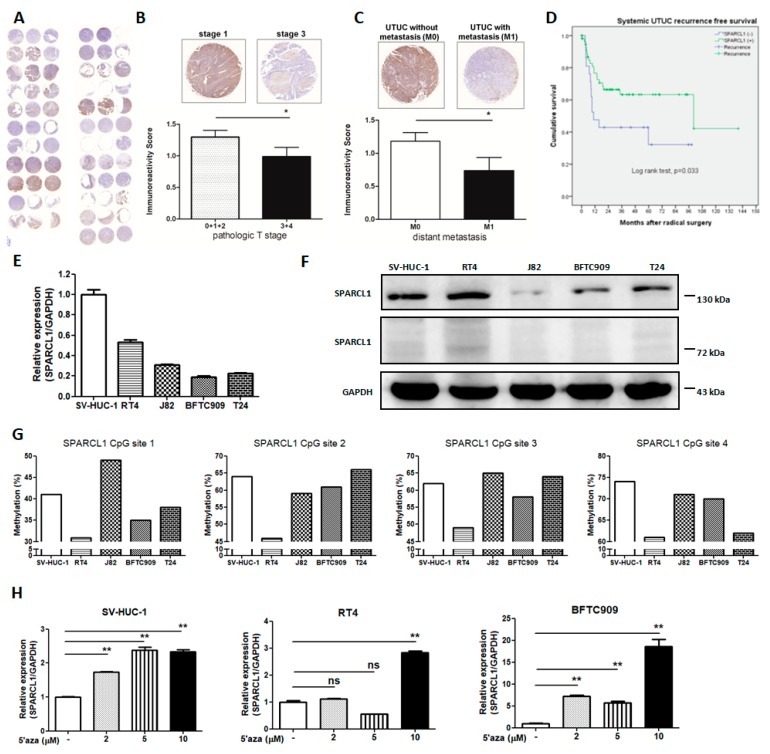
*SPARCL1* was down-regulated in aggressive UTUC via DNA methylation. (**A**) The IHC staining of SPARCL1 in human UTUC TMA samples with triplicate cores per case. (**B**) A significantly reduced expression of SPARCL1 in UTUC with pathologic tumor stages 3 and 4 compared to those with stages 0, 1, and 2 was determined by IHC staining. * *p* < 0.05 between the indicated groups. (**C**) Immunoreactivity score of SPARCL1 protein expression in M0 and M1 tumor of human UTUC. * *p* < 0.05 between the indicated groups. (**D**) Significantly worse systemic UTUC recurrence free survival in SPARCL1 negative UTUCs. (**E**) RT-qPCR and (**F**) Western blots of SPARCL1 expression in an immortalized human urothelial cells (SV-HUC-1), a low grade urothelial carcinoma cell line (RT4) and 3 high grade urothelial carcinoma cell lines (J82, BFTC909, and T24). β-actin was used as a loading control. The predicted molecular weight of SPARCL1 is ~75kDa and we did not detect SPARCL1 monomer bands (~75 kD). SPARCL1 molecular weight was approximately 150 kDa, suggesting that it may be present in the homodimer form. (**G**) *SPARCL1* CpG site methylation levels in UC cell lines. (**H**) The mRNA expression of *SPARCL1* was detected by RT qPCR in SV-HUC-1, RT4 and BFTC909 cell lines treated with 5-aza-2′-deoxycytidine. Error bars represent the mean ± S.E.M., ** *p* < 0.01; ns indicates no significance.

**Figure 4 ijms-20-01653-f004:**
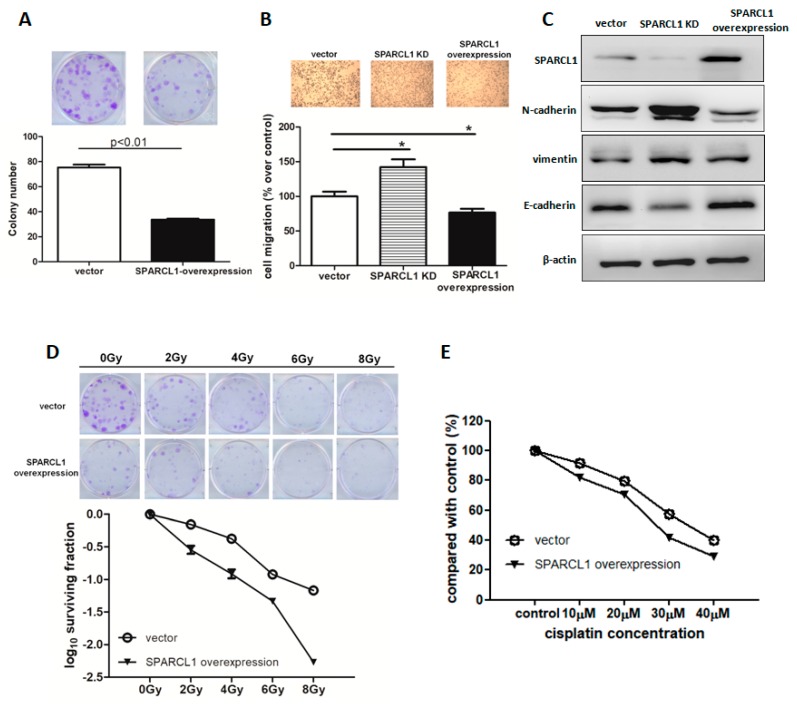
SPARCL1 decreases proliferation and migration of BFTC909 cells and improves the anti-tumor effects of radiation or chemotherapy. (**A**) SPARCL1 overexpression inhibited clonogenicity in vitro. Magnification: 1×. (**B**) Transwell assays were performed to evaluate the effects of SPARCL1 expression on migration of BFTC909 cells. Magnification: 100×. Values are mean ± S.E.M. of three independent experiments. (**C**) SPARCL1 expression is correlated with EMT marker expression. (**D**) SPARCL1 overexpression increased the efficacy of radiotherapy on BFTC909 cells in colony formation assays. Magnification: 1×. (**E**) SPARCL1 overexpression augmented cisplatin effects on cell viability evaluated using MTT assays.

**Table 1 ijms-20-01653-t001:** Patient characteristics.

	SPARCL1 (−) *n* = 22	SPARCL1 (+) *n* = 56	*p* Value
Follow up duration (months)	39.1 ± 30.2	37.8 ± 31.9	0.869
Age (years)	72.6 ± 7.6	68.1 ± 10.9	0.089
Gender(male/female)	8/14	24/32	0.600
Papillary	13	26	0.314
Multifocal	5	10	0.623
LVI	11	25	0.669
CIS	10	17	0.207
SC diff	9	18	0.464
TN	12	19	0.094
ESRD	1	9	0.171
Smoking	4	8	0.668
Prior bladder cancer	1	7	0.297
Systemic recurrence	13	19	0.042

Abbreviation: LVI = Lymphovascular invasion, CIS = Carcinoma in situ, SC diff = Squamous differentiation, TN = Tumor necrosis, ESRD = End stage renal disease.

**Table 2 ijms-20-01653-t002:** Regression analysis for systemic UTUC recurrence.

	Univariate	Multivariate	Hazard Ratio	95% CI
SPARCL1 negative in IHC	0.042	0.011	2.89	1.28~6.54
Non-papillary	0.066	0.008	3.01	1.34~6.76
Gender *	0.070	0.079	-	-
Age **	0.933	0.268	-	-
Multifocal	0.621	0.126	-	-
LVI	0.051	0.147	-	-
CIS	0.970	0.781	-	-
SC diff	0.315	0.617	-	-
TN	0.080	0.360	-	-
ESRD	0.944	0.123	-	-
Smoking	0.185	0.268	-	-
Prior bladder cancer	0.083	0.116	-	-

* = female vs. male, ** = continuous variables.

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
