# Peer review of "Methylation of SPARCL1 Is Associated with Oncologic Outcome of Advanced Upper Urinary Tract Urothelial Carcinoma"

_ijms, 2019, doi:10.3390/ijms20071653_

Round 1

Reviewer 1 Report

The manuscript provides a solid link  between the SPARCL1 methylation status and characteristics of UTUC. Authors performed solid research and support their correlational observations in patients with cell biology studies. The manuscript is well written and focused. Hence I have only few points to rise:

Major.

1.      Comparing the methylation between different cancer cell lines and conclusion are rather speculative - authors should revise this part of results section and discussion accordingly

2.      GAPDH is not proper control for tumor samples - these gene expression is dependent on glucose/metabolic state

Minor

3.      Gene names and mRNA should be italics

4.      Figure 4 - legend bold

5.      sybyr green primers were used - no info about their validation - melting curves ? published? products sequenced?

6.      please provide antibodies details (dilutions, validation, and brands) in methods

Author Response

Reviewer 1:

The manuscript provides a solid link between the SPARCL1 methylation status and characteristics of UTUC. Authors performed solid research and support their correlational observations in patients with cell biology studies. The manuscript is well written and focused. Hence I have only few points to rise:

Major.

1.        Comparing the methylation between different cancer cell lines and conclusion are rather speculative - authors should revise this part of results section and discussion accordingly

Reply:

    Thanks for your comment. The histology of UTUC is urothelial carcinoma. The included cell lines RT4/J82/T24 are bladder urothelial carcinomas. However, we can only obtained the aggressive UTUC cell line (BFTC 909) from Taiwan instead of non-aggressive UTUC cell line (actually all the available UTUC cell lines in the world are aggressive in behavior, which is compatible with the finding from clinical practice). Therefore, we examined other UC cell lines to identify the SPARCL methylation in aggressive UC cell line (BFTC 909/T24/J82) and relatively non-aggressive UC cell line(RT4). (page 11)

    This is the limitation of this study and we mentioned about this in the result section of the revised manuscript (Page 9). This study majorly focus on the SPARCL methylation of UTUC and the role of SPARCL methylation should be further identify in urinary bladder cancer cell line and clinical cohort.

2.        GAPDH is not proper control for tumor samples - these gene expression is dependent on glucose/metabolic state

Reply:

Thanks for your recommendation. This is a very helpful suggestion that we should modify our experimental design for further study. For the experimental design, we initially reviewed the literatures, no reports indicated about direct impact of SPARCL1 on the glucose/metabolic state. Therefore, we used GADPH as control for tumor samples with references as below:

1. SPARCL1 suppresses osteosarcoma metastasis and recruit macrophages by activation of canonical WNT/β-catenin signaling through stabilization of the WNT–receptor complex. Oncogene. 2018 Feb 22;37(8):1049-1061. (GAPDH as reference gene by using QPCR.)
2. Secreted protein acidic and rich in cysteines-like 1 suppresses aggressiveness and predicts better survival in colorectal cancers. Clin Cancer Res. 2012 Oct 1;18(19):5438-48.
3. Secreted protein, acidic and rich in cysteine-like 1 (SPARCL1) is down regulated in aggressive prostate cancers and is prognostic for poor clinical outcome.
Proc Natl Acad Sci U S A. 2012 Sep 11;109(37):14977-82. (Ref 2 and 3 use GAPDH as loading control in western blotting.)

Minor

3.        Gene names and mRNA should be italics

Reply:
Thank you. We have corrected the error in revised manuscript.

4.        Figure 4 - legend bold

Reply:
Thank you. We have corrected the legend in figure in bold in revised manuscript.

5.        sybyr green primers were used - no info about their validation - melting curves ? published? products sequenced?

Reply:
Thanks for your comment. The detailed information is as below:

All primers were purchased from OriGene and checked for specificity using BLAST (NCBI). Exon/intron junctions were spanned.

6.      please provide antibodies details (dilutions, validation, and brands) in methods

Reply: Thanks for your suggestion and we have added the details in methods. (page 11)

Reviewer 2 Report

This manuscript examines expression and potential functions of SPARCL1 in upper urinary tract urothelial carcinomas (UTUC) using a combination of cell line studies and human specimens. Findings of the study are interesting and indicate a potential clinical utility of SPARC1 that would require extensive further investigations beyond the scope of this report. A number of aspects of the data and manuscript were difficult to follow due to inadequate (abbreviated) descriptions of methods or cohorts and problems with English grammar and word usage. However in general I feel that the manuscript is worthy of publication pending amendment of both major and smaller issues as it presents novel results pertaining to this rare tumour type and adds to previous studies of the role of SPARC1 in multiple tumour types.

From the outset, the manuscript was confusing as the Abstract states that “SPARCL1 was the most significantly hypermethylated gene in UTUC” (page 1, lines 36-37) and then states that “RNA level (sic) also revealed significantly more SPARCL1 in UTUC” (page 1, line 39). Gene hypermethylation is more commonly associated with reduced gene transcription and therefore the higher (not lower) mRNA levels reported in the Abstract were puzzling, especially as this apparent anomaly was not discussed. Reading through the manuscript (esp. Figure 2E), it appears that there is a major error in the Abstract and that the statement is incorrect. The statement requires amendment.

There are other instances of similar inaccuracies in the manuscript, for example, on page 3, lines 18-19, the authors state that “SPARCL1 hypermethylation and SPARCL1 mRNA expression was (sic) decreased in UTUC”. The results presented indicate that SPARCL1 hypermethylation was increased, while SPARCL1 mRNA expression was decreased. These types of errors will be appropriately corrected only if English language editing is performed by an experienced scientist who will specifically check that the text matches the data presented in the figures.

There are insufficient details in the Methods. While a thorough revision of all methods will be required, the following examples are given. (a) The authors state that 78 patients were included in the study (page 9, lines 31-32), however 103 specimens were included in the tissue microarray (page 10, line 11), 6 were included in DNA methylation analysis (section 4.2), 25 appear to have been used for DNA methylation validation (Figure 2D) and 55 were used for SPARCL1 mRNA expression (Figure 2E). Proper revision of all numbers, how samples are related to each other and to clinical outcomes, patient details, tumour characteristics and consent procedures is required. (b) References for DNA Methylation Analysis (section 4.2) should be inserted. The statement “normal urothelium of low-grade and low-stage UTUC” is illogical. Suggested alternative “normal urothelium adjacent to low-grade and low-stage UTUC” (if this is what the samples were). (c) The description of bisulphite conversion/pyrosequencing (page 9, lines 46-47) also requires further details or a reference. Note that extensive details are not required, however where several alternative methods are available, the method used should be clear. (d) Immunostaining is not performed on a microscope (page 10, lines 13-14), descriptions of the immunostaining procedure are inadequate or missing (source/dilution of antibodies, duration of incubations, etc should be added) and details of the scoring of immunostained slides are also insufficient. There is no mention (methods or results) of the intracellular localisation of immunoreactivity or intensity of immunostaining and the statement “The identification of SPARCL1 expression was performed by a uropathologist in our institution” is inadequate. For research (reporting of immunostaining that is not used in routine pathology practice and as such does not have validated guidelines), independent scoring and consensus  by at least 2 researchers/pathologists is required, and it could  be considered that if specialist information is a major/pivotal component of the research, that the person or people who specifically performed this work are listed as authors on the manuscript (and thus accountable for the results) or at least acknowledged. (e) Western blotting details are inadequate and require amendment (“The membranes were probed with specific antibodies”). (f) Statements in the description of colony forming assays (section 4.8) are not clear and should be re-written, in particular “plated in 6-well plates based on the stringency of the treatment” and “the number of cells was adjusted to generate 50-200 colonies per well at each radiation dose”. (“SPARCL1 overexpression cells” should be “SPARCL1 overexpressing cells”).

The cell lines used are not properly introduced – the rationale for choice of these cell lines is not clear and the only description of the cell lines is in the legend for Figure 3, rather than the Methods section or the Results section where the cell lines are first mentioned.

The DNA sequence depicted in Figure 2A that includes the 4 sites proposed by the authors to be significantly methylated in UTUC does not appear to be a “CpG island” (as stated by the authors in the Abstract and elsewhere in the manuscript). What definition of CpG island are the authors using? This should be discussed.

In Figures 2C/2D and 2E, were the tumour samples used for DNA methylation analysis (n=25) or SPARCL1 mRNA expression (n=55) high grade or low grade tumours? This should be stated and if there was a mixture, symbols of different colours should be used in the figures and the relevant results discussed in the text.

The immunostaining images are not suitable for publication. These images should be replaced with representative images that clearly depict relevant features of the range of immunostaining profiles detected. Magnification bars should be included in images.

In Figure 3F, the supposed presence of a ~75kD band (in any of the cell lines) is not convincing. The authors should carefully examine whether the antibody that they are using is specific for SPARCL1 and provide a more complete description of the band(s) that they are detecting in Western blots in relation to the predicted ~75kD molecular size of SPARC1 protein (insertion of references would be appropriate).

Error bars are missing from Figure 4D (“vector”) and 4E.

Statements/section titles such as “Low SPARC1 Expression as Associated with Poor Clinical Outcome Owing to Epigenetic Promoter DNA Hypermethylation” (page 4, lines 9-10) are unwarranted based on the experiments performed and should be modified to reflect the methods used and results obtained.

Page 8, line 31: The “prospective” nature of tumour collection is not described in the text. This should be added to the Methods section.

Page 8, line 34: The word “tumorigenesis” refers to the formation of tumours. It is not clear why the authors are proposing that SPARC1 is involved with UTUC “formation” in particular. This should be explained. If the authors are wishing to combine their own results with previously published findings, for example demonstrating that maintenance of SPARC1 expression or SPARC1 overexpression inhibits UTUC formation, these references should be added.

The terms “SPARC1 positive” and “SPARC1 negative” that are used throughout the document are unclear (potentially due to the lack of proper descriptions of SPARC1 mRNA levels or SPARC1 immunohistochemical staining features. For example, in Figure 2E, it appears that only 2 of the 55 tumour cases registered any SPARC1 mRNA expression. Is this true? Then in Figure 3D, where the terms “SPARC1-“ and “SPARC1+” are used again, this is presumably in reference to SPARC1 immunohistochemical staining (<10% tumour cells SPARC1-immunopositive)? The numbers of tumours/patients in each group should be added to the figure to clarify results (and similarly, the numbers of tumours in each group depicted in Figures 3B and 3C should also be added to the graphs). Note that part of the confusion in interpretation of results stems from the different numbers of samples used in mRNA/immunohistochemistry experiments and lack of information regarding how these samples are related to each other or to clinical information.

Page 8, lines 35-36: It is not clear which of the reported investigations were used to derive the recurrence-free survival rates of 60% and 35%.

Page 8, lines 47-48: The conclusions of the authors regarding use of SPARC1 expression as a prognostic biomarker are unwarranted based on the experiments and analyses performed or reported. This statement should be modified to more accurately reflect either the available data or how the current data may be used in the future to evaluate SPARC1 as a biomarker.

Page 8, line 52: There appears to be a phrase missing here “… indicates the important role of tumorigenesis”. Similar comment for the sentence on Page 9, lines 16-17.

Throughout the text, please note that “patients” are not SPARC1 positive or SPARC1 negative, but “UTUC” or “tumours” can be SPARC1 positive or SPARC1 negative.

Author Response

Reviewer 2:

This manuscript examines expression and potential functions of SPARCL1 in upper urinary tract urothelial carcinomas (UTUC) using a combination of cell line studies and human specimens. Findings of the study are interesting and indicate a potential clinical utility of SPARC1 that would require extensive further investigations beyond the scope of this report. A number of aspects of the data and manuscript were difficult to follow due to inadequate (abbreviated) descriptions of methods or cohorts and problems with English grammar and word usage. However in general I feel that the manuscript is worthy of publication pending amendment of both major and smaller issues as it presents novel results pertaining to this rare tumour type and adds to previous studies of the role of SPARC1 in multiple tumour types.

From the outset, the manuscript was confusing as the Abstract states that “SPARCL1 was the most significantly hypermethylated gene in UTUC” (page 1, lines 36-37) and then states that “RNA level (sic) also revealed significantly more SPARCL1 in UTUC” (page 1, line 39). Gene hypermethylation is more commonly associated with reduced gene transcription and therefore the higher (not lower) mRNA levels reported in the Abstract were puzzling, especially as this apparent anomaly was not discussed. Reading through the manuscript (esp. Figure 2E), it appears that there is a major error in the Abstract and that the statement is incorrect. The statement requires amendment.

Reply:

Thanks for your kindly suggestion and we have corrected the errors about RNA level statement in our abstract.

There are other instances of similar inaccuracies in the manuscript, for example, on page 3, lines 18-19, the authors state that “SPARCL1 hypermethylation and SPARCL1 mRNA expression was (sic) decreased in UTUC”. The results presented indicate that SPARCL1 hypermethylation was increased, while SPARCL1 mRNA expression was decreased. These types of errors will be appropriately corrected only if English language editing is performed by an experienced scientist who will specifically check that the text matches the data presented in the figures.

Reply:

Thanks for your kindly suggestion and we have corrected the errors by English editing from our institutional scientists consultation.

There are insufficient details in the Methods. While a thorough revision of all methods will be required, the following examples are given. (a) The authors state that 78 patients were included in the study (page 9, lines 31-32), however 103 specimens were included in the tissue microarray (page 10, line 11), 6 were included in DNA methylation analysis (section 4.2), 25 appear to have been used for DNA methylation validation (Figure 2D) and 55 were used for SPARCL1 mRNA expression (Figure 2E). Proper revision of all numbers, how samples are related to each other and to clinical outcomes, patient details, tumour characteristics and consent procedures is required.

Reply:

Thanks for your comment. We have described the study patient group and its number in revised manuscript. First, we included 6 patients for DNA methylation analysis. Then 25 patients for DNA methylation validation(pyrosequencing) from prospective collected sample since 2016 (the patient numbering of our database is as below). Furhter the SPARCL1 mRNA expression is also from the prospective collected sample since 2016 (the patient numbering of our database is as below). The relationship between SPARCL1 expression and pathological stage / distant metastasis in whole stage distribution TMA cohort (not full slide but only TMA from representative focus by pathologist review). However, for clinical translation, we want to identify the clinical prognostic role of SPARCL1 for advanced UTUC. Therefore, the IHC information from 78 patients for clinical advanced UTUC outcome cohort is from full representative slide by pathologist’s review. (page 9)

DNA methylation (25): 13, 15, 20, 22, 24, 39, 43, 58, 76, 80, 105, 110, 123, 139, 154, 155, 162, 179, 184, 185, 186, 188, 190, 195, 197

mRNA expression (55): 3, 6, 11, 13, 15, 18, 20, 22, 23, 24, 27, 31, 33, 36, 39, 40, 43, 51, 52, 58, 69, 76, 80, 81, 85, 105, 110, 112, 119, 123, 131, 135, 139, 143, 148, 150,51, 154, 155, 160, 165, 167, 170, 171, 172, 174, 176, 178, 179, 180, 184, 185, 188, 189, 192.

(b) References for DNA Methylation Analysis (section 4.2) should be inserted. The statement “normal urothelium of low-grade and low-stage UTUC” is illogical. Suggested alternative “normal urothelium adjacent to low-grade and low-stage UTUC” (if this is what the samples were).

Reply:

Thanks for your comment and we have corrected the description in our revised manuscript.

(c) The description of bisulphite conversion/pyrosequencing (page 9, lines 46-47) also requires further details or a reference. Note that extensive details are not required, however where several alternative methods are available, the method used should be clear.

Reply:

Thanks for your suggestion. We have added more detailed description of bisulphite conversion/pyrosequencing in our revised manuscript.

(d) Immunostaining is not performed on a microscope (page 10, lines 13-14), descriptions of the immunostaining procedure are inadequate or missing (source/dilution of antibodies, duration of incubations, etc should be added) and details of the scoring of immunostained slides are also insufficient. There is no mention (methods or results) of the intracellular localisation of immunoreactivity or intensity of immunostaining and the statement “The identification of SPARCL1 expression was performed by a uropathologist in our institution” is inadequate. For research (reporting of immunostaining that is not used in routine pathology practice and as such does not have validated guidelines), independent scoring and consensus by at least 2 researchers/pathologists is required, and it could  be considered that if specialist information is a major/pivotal component of the research, that the person or people who specifically performed this work are listed as authors on the manuscript (and thus accountable for the results) or at least acknowledged.

Reply:

Thanks for your helpful suggestion in adequate method writing. We have corrected the immunostaining method and detailed procedure in our revised manuscript. The uropathologist (Dr. Min-Tse Sung) and urooncologist (Dr. Hao Lun Luo) in the author list reviewed the slide independently. (page 10)

(e) Western blotting details are inadequate and require amendment (“The membranes were probed with specific antibodies”).

Reply:

Thanks for your helpful suggestion in adequate method writing. We have added the information in the revised manuscript. (page 11)

(f) Statements in the description of colony forming assays (section 4.8) are not clear and should be re-written, in particular “plated in 6-well plates based on the stringency of the treatment” and “the number of cells was adjusted to generate 50-200 colonies per well at each radiation dose”. (“SPARCL1 overexpression cells” should be “SPARCL1 overexpressing cells”).

Reply:

Thanks for your helpful suggestion in adequate method writing. We have corrected the information in the revised manuscript.(page 11)

The cell lines used are not properly introduced – the rationale for choice of these cell lines is not clear and the only description of the cell lines is in the legend for Figure 3, rather than the Methods section or the Results section where the cell lines are first mentioned.

Reply:

The histology of UTUC is urothelial carcinoma. The included cell lines RT4/J82/T24 are bladder urothelial carcinomas. However, we can only obtained the aggressive UTUC cell line (BFTC 909) from Taiwan instead of non-aggressive UTUC cell line (actually all the available UTUC cell lines in the world are aggressive in behavior, which is compatible with the finding from clinical practice). Therefore, we examined other UC cell lines to identify the SPARCL methylation in aggressive UC cell line (BFTC 909/T24/J82) and relatively non-aggressive UC cell line(RT4). (page 11)

The DNA sequence depicted in Figure 2A that includes the 4 sites proposed by the authors to be significantly methylated in UTUC does not appear to be a “CpG island” (as stated by the authors in the Abstract and elsewhere in the manuscript). What definition of CpG island are the authors using? This should be discussed.

Reply:

Thanks for your comment. We have corrected the term of CPG island to CPG site in our revised manuscript.

In Figures 2C/2D and 2E, were the tumour samples used for DNA methylation analysis (n=25) or SPARCL1 mRNA expression (n=55) high grade or low grade tumours? This should be stated and if there was a mixture, symbols of different colours should be used in the figures and the relevant results discussed in the text.

Reply:

Thanks for your comment. All the tumour smaples used for DNA methylation analysis (n=25) or SPARCL1 mRNA expression (n=55) are high grade tumours (clinical significance for further treatment)

The immunostaining images are not suitable for publication. These images should be replaced with representative images that clearly depict relevant features of the range of immunostaining profiles detected. Magnification bars should be included in images.

Reply:

Thanks for your comment. The images have been replaced with representative images as your suggestion.

In Figure 3F, the supposed presence of a ~75kD band (in any of the cell lines) is not convincing. The authors should carefully examine whether the antibody that they are using is specific for SPARCL1 and provide a more complete description of the band(s) that they are detecting in Western blots in relation to the predicted ~75kD molecular size of SPARC1 protein (insertion of references would be appropriate).

Reply:

Thanks for your comment. We did not detect presence of 75kD. SPARCL was reported to presented 150kD as homodimer.

Error bars are missing from Figure 4D (“vector”) and 4E.

Reply:

Thanks for your comment.  The raw data are as below and error bar was too small to present in figure.

Statements/section titles such as “Low SPARC1 Expression as Associated with Poor Clinical Outcome Owing to Epigenetic Promoter DNA Hypermethylation” (page 4, lines 9-10) are unwarranted based on the experiments performed and should be modified to reflect the methods used and results obtained.

Reply:

Thanks for your suggestion. We have corrected the titles to “Low SPARCL1 Expression Is Associated with advanced UTUC stage and more distant metastasis in retrospective TMA cohort

Page 8, line 31: The “prospective” nature of tumour collection is not described in the text. This should be added to the Methods section.

Reply:

Thanks for your comment and we add the description in our revised manuscript. (page 9)

Page 8, line 34: The word “tumorigenesis” refers to the formation of tumours. It is not clear why the authors are proposing that SPARC1 is involved with UTUC “formation” in particular. This should be explained. If the authors are wishing to combine their own results with previously published findings, for example demonstrating that maintenance of SPARC1 expression or SPARC1 overexpression inhibits UTUC formation, these references should be added.

Reply:

Thanks for your comment. We agree with your suggestion and the reviewer 3 also suggest us to remove this in discussion session.

The terms “SPARC1 positive” and “SPARC1 negative” that are used throughout the document are unclear (potentially due to the lack of proper descriptions of SPARC1 mRNA levels or SPARC1 immunohistochemical staining features. For example, in Figure 2E, it appears that only 2 of the 55 tumour cases registered any SPARC1 mRNA expression. Is this true? Then in Figure 3D, where the terms “SPARC1-“ and “SPARC1+” are used again, this is presumably in reference to SPARC1 immunohistochemical staining (<10% tumour cells SPARC1-immunopositive)? The numbers of tumours/patients in each group should be added to the figure to clarify results (and similarly, the numbers of tumours in each group depicted in Figures 3B and 3C should also be added to the graphs). Note that part of the confusion in interpretation of results stems from the different numbers of samples used in mRNA/immunohistochemistry experiments and lack of information regarding how these samples are related to each other or to clinical information.

Reply:

Page 8, lines 35-36: It is not clear which of the reported investigations were used to derive the recurrence-free survival rates of 60% and 35%.

Reply:

Thanks for your comment and we have added the description “presented as Kaplan Meier plot in Fig-3D” in page 8

Page 8, lines 47-48: The conclusions of the authors regarding use of SPARC1 expression as a prognostic biomarker are unwarranted based on the experiments and analyses performed or reported. This statement should be modified to more accurately reflect either the available data or how the current data may be used in the future to evaluate SPARC1 as a biomarker.

Reply:

Thanks for you comment. From the locally advanced UTUC cohort in Fig-3D. The positive IHC is associated with better recurrence free survival. In addition, the sensitivity if SPARCL1 overexpressing UTUC cell line is more sensitive to cradiation and chemotherapy. We have corrected our description to “SPARCL1 could be considered a prognostic biomarker for advanced UTUC and further identification of the outcome of the clinical benefit of chemotherapy or radiation on patients with positive SPARCL1 UTUC needs further clinical trial for validation of its prognostic utility.” to reflect the direction of future effort.

Page 8, line 52: There appears to be a phrase missing here “… indicates the important role of tumorigenesis”. Similar comment for the sentence on Page 9, lines 16-17.

Reply:

Thanks for your comment. We have revised to “loss of SPARCL1 function might be considered to have clinical utility in assess UTUC cancer behaviour”

Throughout the text, please note that “patients” are not SPARC1 positive or SPARC1 negative, but “UTUC” or “tumours” can be SPARC1 positive or SPARC1 negative.

Reply:

Thanks for your comment. We have corrected our description according to your suggestion.

Reviewer 3 Report

In this manuscript, the authors showed that the gene encoding the secreted protein acidic rich in cysteine-like 1 (SPARCL1) is hypermethylated in high-stage/high-grade UTUC tumour samples compared with normal urothelium of low-stage/low-grade specimens. Additionally, in vitro studies revealed that the SPARCL1 overexpression in BFTC-909 cells induced a less aggressive behavior and increased the sensitive to radiation chemotherapy. The quality of the manuscript is apparently good.

Introduction section could be improved with additional evidence about urothelial carcinoma and/or SPARCL1.  This sentence should be reported in the conclusion or discussion section “We found that high-stage/high-grade UTUC tumour samples had significant SPARCL1 hypermethylation compared with normal urothelium of low-stage/low-grade specimens”.

 I think that lines 32-36 should be moved in discussion section.

What is the meaning of C and  T? I suppose that is control and tumor. Please add abbreviation.

The following sentence should be reported in discussion section “The previous report showed  SPARCL1 suppresses the proliferation and migration of human ovarian cancer [13].”

Please add abbreviation list.

Author Response

Reviewer 3:

In this manuscript, the authors showed that the gene encoding the secreted protein acidic rich in cysteine-like 1 (SPARCL1) is hypermethylated in high-stage/high-grade UTUC tumour samples compared with normal urothelium of low-stage/low-grade specimens. Additionally, in vitro studies revealed that the SPARCL1 overexpression in BFTC-909 cells induced a less aggressive behavior and increased the sensitive to radiation chemotherapy. The quality of the manuscript is apparently good.

Introduction section could be improved with additional evidence about urothelial carcinoma and/or SPARCL1.  This sentence should be reported in the conclusion or discussion section “We found that high-stage/high-grade UTUC tumour samples had significant SPARCL1 hypermethylation compared with normal urothelium of low-stage/low-grade specimens”.

Reply:

Thanks for your comment. We have revised out manuscript according to your comments especially in conclsion an add some description about potential clinical utility in our discussion.

 I think that lines 32-36 should be removed in discussion section.

Reply:

Thanks for your comment. We agree with you and reviewer 2 also suggested us to remove this description in discussion section.

What is the meaning of C and T? I suppose that is control and tumor. Please add abbreviation.

Reply:

Thanks for your suggestion. We have corrected the abbreviation by replacement of C and T to control and toumor in Fig-2.

The following sentence should be reported in discussion section “The previous report showed SPARCL1 suppresses the proliferation and migration of human ovarian cancer [13].”

Reply:

Thanks for your comment. We have described the reported mechanism in page 9.

Please add abbreviation list.

Reply:

Thanks for your suggestion. We have added abbreviation list in the abstract.

Round 2

Reviewer 1 Report

all my concerns were addresed

Author Response

Thanks for your recommendation and suggestion again. In order to improve the quality of this manuscript. We have revised the manuscript in terms of English language and scientific writing style by MDPI English editing service before acceptance. (English ID: english-8722).

Reviewer 2 Report

The authors have addressed many of the issues raised by reviewers and this has considerably improved the manuscript in terms of both the level of detail in descriptions of methods and results, and the ease with which the text can be read and understood. However, many issues remain in term of English language (grammar, word usage) and scientific (as opposed to conversational) writing style. There are a number of spelling errors throughout the document and several of the reviewers’ comments that the authors state have been corrected are not included in the manuscript copy provided for re-review. For example, representative images of SPARC1 immunostaining that the authors state to be inserted in place of the low power (indistinct) images are missing, descriptions of the intracellular localisation of SPARC1 in the tissue samples is not described, and although the authors state that they did not detect SPARC1 monomer bands (~75kD) in western blots, the Figure 3 legend implies that monomer bands were detected. These are just examples of many errors that require correction. There are many more errors that are too numerous to mention and beyond the reasonable scope of a scientific review. I still feel that the manuscript is worthy of publication, but only if the authors more rigorously address scientific and English language editing of the text and figures.

Note that details regarding the origins of all tissue samples and how these relate to each other in the various parts of the investigations are critical for evaluation of results of the study (for readers of the published manuscript). These need to be inserted into the manuscript in a systematic and scientific manner, possibly as Supplementary Data. This includes tissue samples included in the tissue microarray. In the manuscript, is not clear whether DNA, RNA and immunohistochemistry studies were performed using the same samples or on independent samples, although there does appear to be some overlap based on the response to reviewers’ comments. In addition, the text describing Figures 3A, 3B and 3C implies that the data were all derived from the tissue microarrays, but it appears that this was not the case. Insertion of numbers represented in each group would clarify some of these issues. It is noted that the authors did not “Reply” to this comment from the previous review and they have possibly submitted the response without completing revisions.

Author Response

The authors have addressed many of the issues raised by reviewers and this has considerably improved the manuscript in terms of both the level of detail in descriptions of methods and results, and the ease with which the text can be read and understood. However, many issues remain in term of English language (grammar, word usage) and scientific (as opposed to conversational) writing style. There are a number of spelling errors throughout the document.

Answer:

Thanks for your valuable suggestion. The manuscript was revised on March 18th by MDPI English editing service (English ID: english-8722).

Several of the reviewers’ comments that the authors state have been corrected are not included in the manuscript copy provided for re-review. For example, representative images of SPARC1 immunostaining that the authors state to be inserted in place of the low power (indistinct) images are missing, descriptions of the intracellular localisation of SPARC1 in the tissue samples is not described, and although the authors state that they did not detect SPARC1 monomer bands (~75kD) in western blots, the Figure 3 legend implies that monomer bands were detected. These are just examples of many errors that require correction.

Answer:

Thanks for your comment. We have modified the following passages. We described the positive SPARCL1 staining was mainly located in the cytoplasm (Supplementary Figure 1). For the legend of Figure 3, our previous description may mislead the reader and we correct our description in this part that we did not detect the monomer bands (~75kD) in the revised version.

There are many more errors that are too numerous to mention and beyond the reasonable scope of a scientific review. I still feel that the manuscript is worthy of publication, but only if the authors more rigorously address scientific and English language editing of the text and figures.

Answer:

Thanks for your recommendation to improve our manuscript. We revised this manuscript again by MDPI English editing service (English ID: english-8722).

Note that details regarding the origins of all tissue samples and how these relate to each other in the various parts of the investigations are critical for evaluation of results of the study (for readers of the published manuscript). These need to be inserted into the manuscript in a systematic and scientific manner, possibly as Supplementary Data. This includes tissue samples included in the tissue microarray. In the manuscript, is not clear whether DNA, RNA and immunohistochemistry studies were performed using the same samples or on independent samples, although there does appear to be some overlap based on the response to reviewers’ comments. In addition, the text describing Figures 3A, 3B and 3C implies that the data were all derived from the tissue microarrays, but it appears that this was not the case. Insertion of numbers represented in each group would clarify some of these issues.

Answer: Thanks for your comments. The TMA numbering/DNA/RNA/IHC for advanced cohort was attached in the supplement file. The Supplement File 2 includes the detail of TMA, DNA, and RNA characteristics for analysis (Excel file). When isolating RNA and DNA from the adjacent urothelium tissue samples, it is sometimes difficult to extract enough concentration and quality for DNA methylation and QPCR experiments because the amount of the sample is too small. We honestly present our tissue sample numbering and detailed clinical information in the Supplement File 2.

It is noted that the authors did not “Reply” to this comment from the previous review and they have possibly submitted the response without completing revisions.

Answer: Thanks for your comments to make this work more readable and more scientific. In this updated version of our manuscript, we have revised the manuscript according to your recommendation point by point as we can.

Round 3

Reviewer 2 Report

I do not feel that the entire manuscript has undergone English language editing as there are still numerous errors, some of which I have listed below, along with a few remaining questions. The authors have addressed some of the questions raised during review. Spelling in the manuscript is a mixture of UK and US English – this should be made consistent.

This comment is in “plain English” and not scientific language. ‘UTUC’ stands for ‘urinary tract urothelial carcinoma’. ‘Carcinoma’ is a type of tumour. As such, writing ‘UTUC tumor’ is incorrect as a scientist or clinician wouldn’t say ‘carcinoma tumors’ to describe a tumour, it is the same as saying ‘tumor tumors’. This error should be corrected throughout the document.

Abstract line 6: ‘was analyzed’ should be ‘were analyzed’

Abstract line 7: ‘UTUC tissues’

Abstract lines 7-10: ‘Then we prospectively collected UTUC samples and adjacent normal urothelium for pyrosequencing validation, identifying significant CpG site methylation in UTUC tissues. In addition, SPARCL1 RNA levels were significantly lower in UTUC samples.’

Abstract line 10: The term ‘retrospectively reviewed’ is not relevant in this context and can be removed.

Abstract line 11: ‘carcinoma’ should be ‘carcinomas’

Introduction line 10: “Many literatures” could be “Several studies”

Introduction line 15: The definition of a “tumour suppressor” is NOT that it induces cell differentiation. This sentence should be rewritten.

Introduction line 17: “UTUC tumour” should be “UTUC”

Introduction line 18: “urothelium of” should be “urothelium adjacent to”

Section 2.1, line 1: delete the word “the” (x2)

Section 2.1, line 5, “were” should be “are”

Section 2.1, line 7: “based on differences in DNA methylation”

Section 2.1, lines 7-9: The top 10 hypermethylated genes are shown in Figure 1c; SPARCL1 hypermethylation was found to be increased 20-fold in UTUC tissue (p=4x10-5).

Section 2.2, lines 1-2: delete “and the DNA methylation status of SPARCL1 were (sic) visualized in Figure 2A”. Insert (Figure 2A) at the end of the following sentence.

Section 2.2, line 5: “ was shown”  should be “is shown”

Section 2.2, line 6: “CpG sites hypermethylation” should be “CpG site hypermethylation” (In English, adjectives are always singular)

Figure 1 legend, lines 2-3: “3 normal urothelium samples and the raw data generated was analysed with Partek.”

Figure 1 legend, line 4: “region at” should be “regions of”

Figure 1 legend, lines 4-5: “Principal component analysis plot based on methylation profiles in UTUC (red) and normal urothelium (blue).

Figure 1 legend, line 6: “normal samples” should be “normal urothelium samples”

Figure 2 legend, line 1: “hyper-DMR” should be written in full. “upper urinary tract urothelial carcinoma”

Figure 2 legend, (B): “Four methylation sites in the SPARCL1 promoter region identified by bisulphite pyrosequencing.”

Figure 2 legend, line 5: “urothelium” is a noun, “urothelial” is an adjective, so either “adjacent urothelium” or “adjacent urothelial tissue”

Figure 2 legend, line 6: delete “resulted”

Page 5, line 4: “more commonly developed…” Note that the results of the authors suggest that absence of SPARCL1 expression indicates early recurrence (within 2 years), but after 2 years, recurrence is uncommon and not dependent on presence or absence of SPARCL1 expression in the tumours. This may have important implications for patient management (surveillance).

Figure 3 legend, lines 2 and 6 “significant” should be “significantly”

Figure legend, lines 4 and 6: delete “indicates that”

Figure 3 legend, lines 11-12: This sentence has not been corrected properly. The authors state that they didn’t detect 75kD bands, which means that they did NOT detect bands corresponding in size to SPARCL1 monomers. So they can’t write that “it may be present as monomer as well as homodimer” (sic).

Section 2.4, title: “of Cisplatin”

Section 2.4, line 2: “cell line” should be “cell lines”

Section 2.4, line 3: “had a weak SPARCL1 presentation” should be “weakly expressed SPARCL1,”

Section 2.4, line 4 onwards: “SPARCL1 overexpression was induced in the BFTC909 cell line, following which colony assays were performed, indicating that overexpression of SPARCL1 in vitro significantly decreased cell viability (Figure 4A). A previous report similarly showed that…”

Section 2.4, lines 7-8: “Therefore we investigated whether SPARCL1 plays a role in metastasis of UTUC. The migration of BFTC909 cells was significantly enhanced by SPARCL1 knockdown, while overexpression of SPARCL1 inhibited cell migration (Figure 4B), indicating that…”

Section 2.4, line 12: “has been known” should be “is known”

Section 2.4, line 13: “attempted to examine…” should be “examined the correlation between SPARCL1 expression and expression of EMT markers”

Section 2.4, line 14: delete “led to a”

Section 2.4, line 15: delete “by western blotting analysis”

Section 2.4, line 18: delete “that”

Section 2.4, lines 19, 20: The authors should check that SPARCL1 effects on radiation or cisplatin treatments are “synergistic” (the formulae should be added to the methods section). If the effects are additive and not synergistic, the text should be altered accordingly.

Section 2.4, line 21: “suppresses the proliferation” should be “suppresses proliferation”

Section 2.4, line 22: “with cisplatin treatment” should be “of cisplatin treatment”

Figure 4 legend: “SPARCL1 decreases proliferation and migration of BFTC909 cells and improves the anti-tumour effects of radiation or chemotherapy. (A) SPARCL1 overexpression inhibited clonogenicity in vitro. (B) Transwell assays were performed to evaluate the effects of SPARCL1 expression on migration of BFTC909 cells. Values are mean + S.E.M. of three independent experiments. (C) SPARCL1 expression is correlated with EMT marker expression. (D) SPARCL1 overexpression increased the efficacy of radiotherapy on BFTC909 cells in colony formation assays. (E) SPARCL1 overexpression augmented cisplatin effects on cell viability evaluated using MTT assays.

Page 9, line 9: “curative” should be “curable”

Page 9, paragraph 3, line 8: the phrase “to have clinical utility in assess (sic) UTUC cancer (sic) behaviour” is not English and it is not clear what the authors are referring to. This sentence should be re-written.

Page 9, paragraph 4, line 2: “comparing with” should be “compared to”

Page 9, paragraph 4, line 4: delete “cancer”; “colony assays” should be “colony formation assays”

Page 9, paragraph 4, line 5: “indicated” should be “indicates”

Page 9, paragraph 4, line 8: “overexpression cell line” should be “overexpressing cells”; “study” should be “studies”

Page 10, paragraph 1, lines 3-4: The sentence “The negative association between SPARCL1 expression and poor tumour differentiation indicates the important role of tumorigenesis” does not make sense and needs to be re-written.

Page 10, paragraph 1, line 7: delete the semi-colons (;) and use commas (,). What do the authors mean by “interference” with the microenvironment?

Page 10, paragraph 1, lines 8-9: “found that SPARCL1 methylation was more common”

Page 10, paragraph 2, line 11: delete “cancer”

Page 10, paragraph 2, line 13: Do the authors mean prognostic or predictive here? (prognostic relates to survival)

Page 10, paragraph 2, line 14: “To better understand the methylation status between the aggressive and non-aggressive cell line.” is not a complete sentence and needs to be rewritten.

I have not proofread the Methods section – spelling errors and sentence fragments should be corrected.

Conclusions section, line 1: delete “tumour”

Conclusions section, line 4: delete “cancer”, “more sensitive” should be “increased sensitivity”

Author Response

Reviewer’s comment

I do not feel that the entire manuscript has undergone English language editing as there are still numerous errors, some of which I have listed below, along with a few remaining questions. The authors have addressed some of the questions raised during review. Spelling in the manuscript is a mixture of UK and US English – this should be made consistent.

This comment is in “plain English” and not scientific language. ‘UTUC’ stands for ‘urinary tract urothelial carcinoma’. ‘Carcinoma’ is a type of tumour. As such, writing ‘UTUC tumor’ is incorrect as a scientist or clinician wouldn’t say ‘carcinoma tumors’ to describe a tumour, it is the same as saying ‘tumor tumors’. This error should be corrected throughout the document.

Answer: Thank you for suggestions, which have enhanced the readability and scientific accuracy of the statements in the manuscript. We have corrected the errors regarding UTUC in the revised manuscript.

Abstract line 6: ‘was analyzed’ should be ‘were analyzed’

Abstract line 7: ‘UTUC tissues’

Abstract lines 7-10: ‘Then we prospectively collected UTUC samples and adjacent normal urothelium for pyrosequencing validation, identifying significant CpG site methylation in UTUC tissues. In addition, SPARCL1 RNA levels were significantly lower in UTUC samples.’

Abstract line 10: The term ‘retrospectively reviewed’ is not relevant in this context and can be removed.

Abstract line 11: ‘carcinoma’ should be ‘carcinomas’

Answer: As suggested, we have corrected the abstract point by point according to your recommendation.

Introduction line 10: “Many literatures” could be “Several studies”

Introduction line 15: The definition of a “tumour suppressor” is NOT that it induces cell differentiation. This sentence should be rewritten.

Introduction line 17: “UTUC tumour” should be “UTUC”

Introduction line 18: “urothelium of” should be “urothelium adjacent to”

Answer: Thank you for your comments. We have made relevant revisions in the Introduction section. We have revised the mentioned sentence: “It is also a tumour suppressor as it induces cell differentiation possibly via MET, which represses the aggressiveness of CRCs.”

Section 2.1, line 1: delete the word “the” (x2)

Section 2.1, line 5, “were” should be “are”

Section 2.1, line 7: “based on differences in DNA methylation”

Section 2.1, lines 7-9: The top 10 hypermethylated genes are shown in Figure 1c; SPARCL1hypermethylation was found to be increased 20-fold in UTUC tissue (p=4x10-5).

Section 2.2, lines 1-2: delete “and the DNA methylation status of SPARCL1 were (sic) visualized in Figure 2A”. Insert (Figure 2A) at the end of the following sentence.

Section 2.2, line 5: “ was shown”  should be “is shown”

Section 2.2, line 6: “CpG sites hypermethylation” should be “CpG site hypermethylation” (In English, adjectives are always singular)

Answer: Thank you for your comments. We have made relevant revisions in Section 2.1 and 2.2.

Figure 1 legend, lines 2-3: “3 normal urothelium samples and the raw data generated was analysed with Partek.”

Figure 1 legend, line 4: “region at” should be “regions of”

Figure 1 legend, lines 4-5: “Principal component analysis plot based on methylation profiles in UTUC (red) and normal urothelium (blue).

Figure 1 legend, line 6: “normal samples” should be “normal urothelium samples”

Figure 2 legend, line 1: “hyper-DMR” should be written in full. “upper urinary tract urothelial carcinoma”

Figure 2 legend, (B): “Four methylation sites in the SPARCL1 promoter region identified by bisulphite pyrosequencing.”

Figure 2 legend, line 5: “urothelium” is a noun, “urothelial” is an adjective, so either “adjacent urothelium” or “adjacent urothelial tissue”

Figure 2 legend, line 6: delete “resulted”

Answer: Thank you for your suggestion. We have corrected the legends of Figures 1 and 2. “DMR” has been expanded to “differentially-methylated region” and has been added to the abbreviation list.

Page 5, line 4: “more commonly developed…” Note that the results of the authors suggest that absence of SPARCL1 expression indicates early recurrence (within 2 years), but after 2 years, recurrence is uncommon and not dependent on presence or absence of SPARCL1 expression in the tumours. This may have important implications for patient management (surveillance).

Figure 3 legend, lines 2 and 6 “significant” should be “significantly”

Figure legend, lines 4 and 6: delete “indicates that”

Figure 3 legend, lines 11-12: This sentence has not been corrected properly. The authors state that they didn’t detect 75kD bands, which means that they did NOT detect bands corresponding in size to SPARCL1 monomers. So they can’t write that “it may be present as monomer as well as homodimer” (sic).

Answer: Thank you for your suggestion. We have re-written the legend of Figure 3. We have revised the corresponding description as follows: “SPARCL1 molecular weight was approximately 150 kDa, suggesting that it may be present in the homodimer form.”

Section 2.4, title: “of Cisplatin”

Section 2.4, line 2: “cell line” should be “cell lines”

Section 2.4, line 3: “had a weak SPARCL1 presentation” should be “weakly expressed SPARCL1,”

Section 2.4, line 4 onwards: “SPARCL1 overexpression was induced in the BFTC909 cell line, following which colony assays were performed, indicating that overexpression of SPARCL1 in vitro significantly decreased cell viability (Figure 4A). A previous report similarly showed that…”

Section 2.4, lines 7-8: “Therefore we investigated whether SPARCL1 plays a role in metastasis of UTUC. The migration of BFTC909 cells was significantly enhanced by SPARCL1 knockdown, while overexpression of SPARCL1 inhibited cell migration (Figure 4B), indicating that…”

Section 2.4, line 12: “has been known” should be “is known”

Section 2.4, line 13: “attempted to examine…” should be “examined the correlation between SPARCL1 expression and expression of EMT markers”

Section 2.4, line 14: delete “led to a”

Section 2.4, line 15: delete “by western blotting analysis”

Section 2.4, line 18: delete “that”

Section 2.4, lines 19, 20: The authors should check that SPARCL1 effects on radiation or cisplatin treatments are “synergistic” (the formulae should be added to the methods section). If the effects are additive and not synergistic, the text should be altered accordingly.

Section 2.4, line 21: “suppresses the proliferation” should be “suppresses proliferation”

Section 2.4, line 22: “with cisplatin treatment” should be “of cisplatin treatment”

Answer: Thank you for your suggestions. We have made relevant revisions in Section 2.4. We have revised our description as follows: “We observed that SPARLC1 overexpression enhanced antitumour effect of cisplatin treatment, as presented in Figure 4E.”

Figure 4 legend: “SPARCL1 decreases proliferation and migration of BFTC909 cells and improves the anti-tumour effects of radiation or chemotherapy. (A) SPARCL1 overexpression inhibited clonogenicity in vitro. (B) Transwell assays were performed to evaluate the effects of SPARCL1 expression on migration of BFTC909 cells. Values are mean + S.E.M. of three independent experiments. (C) SPARCL1 expression is correlated with EMT marker expression. (D) SPARCL1 overexpression increased the efficacy of radiotherapy on BFTC909 cells in colony formation assays. (E) SPARCL1 overexpression augmented cisplatin effects on cell viability evaluated using MTT assays.

Answer: Thank you for your suggestion. We have re-written the legend of Figure 4.

Page 9, line 9: “curative” should be “curable”

Page 9, paragraph 3, line 8: the phrase “to have clinical utility in assess (sic) UTUC cancer (sic) behaviour” is not English and it is not clear what the authors are referring to. This sentence should be re-written.

Page 9, paragraph 4, line 2: “comparing with” should be “compared to”

Page 9, paragraph 4, line 4: delete “cancer”; “colony assays” should be “colony formation assays”

Page 9, paragraph 4, line 5: “indicated” should be “indicates”

Page 9, paragraph 4, line 8: “overexpression cell line” should be “overexpressing cells”; “study” should be “studies”

Answer: Thank you for your suggestions. We have made relevant revisions in page 9. We have revised our description as follows: “Such a high prevalence of SPARCL1 hypermethylation in the UTUC samples indicates that the loss of SPARCL1 function might be considered to have clinical utility in the assessment of UTUC behavior.”

Page 10, paragraph 1, lines 3-4: The sentence “The negative association between SPARCL1 expression and poor tumour differentiation indicates the important role of tumorigenesis” does not make sense and needs to be re-written.

Page 10, paragraph 1, line 7: delete the semi-colons (;) and use commas (,). What do the authors mean by “interference” with the microenvironment?

Page 10, paragraph 1, lines 8-9: “found that SPARCL1 methylation was more common”

Page 10, paragraph 2, line 11: delete “cancer”

Page 10, paragraph 2, line 13: Do the authors mean prognostic or predictive here? (prognostic relates to survival)

Page 10, paragraph 2, line 14: “To better understand the methylation status between the aggressive and non-aggressive cell line.” is not a complete sentence and needs to be rewritten.

Answer: Thank you for your suggestions. We have made relevant revisions in page 10. We have revised our description as follows: “The negative association between the SPARCL1 expression and poor tumour differentiation indicates the major role of aggressive tumour behavior.” In page 10, paragraph 1, line 7: We re-wrote “interference with” to “regulation of” that is better word choice referring to SPARCL1 contributes functionally to endothelial cell quiescence and blood vessel homeostasis. In page 10, paragraph2, line 13: our intended meaning is “prognostic”.  In addition, we have revised the description on page 10, paragraph3, line 14 as follows: “To better understand the methylation status between the aggressive and non-aggressive cell lines, the methylation status in primary culture cells should be validated.”

I have not proofread the Methods section – spelling errors and sentence fragments should be corrected.

Conclusions section, line 1: delete “tumour”

Conclusions section, line 4: delete “cancer”, “more sensitive” should be “increased sensitivity”

Answer: Thank you for your suggestions. We have revised the Methods section again as well as the Conclusions section.
